# BMI1 regulates PRC1 architecture and activity through homo- and hetero-oligomerization

Felicia Gray[1,*], Hyo Je Cho[1,*], Shirish Shukla[1,*], Shihan He[1], Ashley Harris[2,3], Bohdan Boytsov[1], Łukasz Jaremko[4,5], Mariusz Jaremko[5], Borries Demeler[6], Elizabeth R. Lawlor[1,2,3], Jolanta Grembecka[1] & Tomasz Cierpicki[1]

BMI1 is a core component of the polycomb repressive complex 1 (PRC1) and emerging data support a role of BMI1 in cancer. The central domain of BMI1 is involved in protein–protein interactions and is essential for its oncogenic activity. Here, we present the structure of BMI1 bound to the polyhomeotic protein PHC2 illustrating that the central domain of BMI1 adopts an ubiquitin-like (UBL) fold and binds PHC2 in a β-hairpin conformation. Unexpectedly, we find that the UBL domain is involved in homo-oligomerization of BMI1. We demonstrate that both the interaction of BMI1 with polyhomeotic proteins and homo-oligomerization via UBL domain are necessary for H2A ubiquitination activity of PRC1 and for clonogenic potential of U2OS cells. Here, we also emphasize need for joint application of NMR spectroscopy and X-ray crystallography to determine the overall structure of the BMI1–PHC2 complex.

[1] Department of Pathology, University of Michigan, Ann Arbor, Michigan 48109, USA. [2] Translational Oncology Program, University of Michigan, Ann Arbor, Michigan 48109, USA. [3] Department of Pediatrics and Communicable Diseases, University of Michigan, Ann Arbor, Michigan 48109, USA. [4] Deutsches Zentrum fur Neurodegenerative Erkrankungen (DZNE), Am Fassberg 11, 37077 Goettingen, Germany. [5] Max-Planck Institute of Biophysical Chemistry, NMR-based Department for Structural Biology, Am Fassberg 11, 37077 Goettingen, Germany. [6] Department of Biochemistry, The University of Texas Health Science Center at San Antonio, San Antonio, Texas 78229, USA. * These authors contributed equally to this work. Correspondence and requests for materials should be addressed to T.C. (email: tomaszc@umich.edu).

BMI1 (B cell-specific Moloney murine leukemia virus integration site 1) is a polycomb group family member and emerging data support an important role for BMI1 in cancer. The gene encoding *BMI1* was initially identified as an oncogene inducing B- and T-cell leukemias[1]. Further studies found that *BMI1* is a stem cell gene that determines the proliferative capacity and self-renewal of normal and leukemic stem cells[2]. BMI1 is frequently overexpressed in patients with hematologic[3–5] and solid cancers[6–8]. Silencing of *BMI1* impairs cancer cell proliferation and tumour growth in cancer models[9–15], suggesting that BMI1 might represent a valid target for therapeutic intervention[16,17].

The mammalian polycomb repressive complex 1 (PRC1) is a multisubunit protein complex involved in gene silencing[18,19]. The canonical PRC1 complex is composed of four core subunits: CBX (polycomb; CBX2/4/6/7/8), PCGF (polycomb group factors; PCGF1–6), PHC (polyhomeotic homologues; PHC1/2/3) and RING E3 ligase (RING1A/B)[18,19]. The presence of numerous orthologs results in diverse compositions of PRC1 with potentially different functions[19–21]. PRC1 has at least two distinct activities contributing to repressed gene transcription: mono-ubiquitination of histone H2A on Lys119 (refs 22,23) and chromatin compaction[24,25]. The BMI1 protein, also known as PCGF4 (polycomb group RING finger protein 4), is a central component of the canonical PRC1 complex and has a dual role in PRC1 activity: regulation of H2A ubiquitination activity[26–28] and mediation of protein–protein interactions[29–33].

BMI1 is a 37 kDa protein composed of three distinct regions: a N-terminal RING domain[26,27], a central domain[34] and a C-terminal proline-serine rich domain involved in the regulation of protein stability[35]. The RING domain of BMI1 forms a complex with RING1A/B proteins, which constitutes the heterodimeric E3 ubiquitin ligase subunit of the PRC1 complex[26,27]. BMI1 itself has no ubiquitin ligase activity but through a direct interaction it stabilizes RING1A/B, leading to increased H2A ubiquitination activity[26,28]. The central domain of BMI1 was initially predicted as a putative helix-turn-helix (HTH) domain[36] and more recently was defined as an ubiquitin-like (UBL) domain, also called RAWUL (RING finger- and WD40-associated ubiquitin-like) domain[34]. This domain is involved in protein–protein interactions and its best characterized binding partners are the polyhomeotic proteins (PHC1, PHC2, PHC3)[29,30]. In addition to interactions within PRC1, the BMI1 central domain has also been implicated in other protein–protein interactions, including the transcription factors E4F1 (ref. 31), Zfp277 (ref. 32) and the PLZF-RARA fusion protein[33].

Functional studies revealed that the central domain of BMI1 is essential for its oncogenic activity. Deletion analysis shows that this domain is necessary for transcriptional repression activity[36], immortalization of mammary epithelial cells[37] and lifespan extension of human fibroblasts[38]. However, the structure and molecular mechanisms determining how the central domain of BMI1 contributes to the overall architecture and function of the canonical PRC1 complex have not been fully elucidated. To address these questions we determined the three-dimensional structure of the PHC2–BMI1 complex revealing that the BMI1 central domain adopts an ubiquitin-like (UBL) fold and binds a short, 24 amino acid fragment of PHC2 in a β-hairpin conformation. Unexpectedly, we find that the UBL domain is involved in homo-oligomerization of BMI1. Our work reveals that both hetero- and homo-oligomerization of the UBL domain contribute to BMI1 function and activity.

## Results

### The BMI1 central domain binds directly to the PHC2 HD1.

The central domain of BMI1 has been reported to interact with the polyhomeotic PHC2 protein and we sought to characterize the molecular details of this interaction (Fig. 1a)[29,30]. To confirm a direct interaction between BMI1 and PHC2 we developed a cellular pulldown assay overexpressing an Avi-tagged BMI1 construct lacking the N-terminal RING domain (Avi-BMI1$_{106-326}$) and a Myc-tagged full-length short isoform of PHC2 (PHC2_B). Streptavidin pulldown of biotinylated Avi-BMI1$_{106-326}$ demonstrates that the fragment of BMI1 lacking the RING domain interacts with Myc-PHC2 (Fig. 1b). Further co-immunoprecipitation experiments confirmed the interaction of full length BMI1 with PHC2 (Supplementary Fig. 1a) and showed slightly less efficient binding with full length BMI1 when compared to BMI1$_{106-326}$ (Supplementary Fig. 1b). To define the central domain within BMI1 that binds PHC2 and is suitable for structural studies we employed bioinformatic analysis and selected a construct encompassing BMI1 residues 106–240. While the $^1$H–$^{15}$N HSQC spectrum for this construct is consistent with a folded domain, it is not optimal for structural studies due to the presence of a significant number of disordered residues (Fig. 1c, top). To define the boundaries of the globular central domain we employed carbon detected NMR experiments to efficiently identify flexible regions. We assigned disordered fragments in BMI1$_{106-240}$ by employing a combination of $^{13}$C-detected 2D CACO, CBCACO and CANCO experiments using a previously published protocol[39]. We identified residues 106–120 and 236–240 as being highly flexible in solution (Supplementary Fig. 2). Deletion of these residues significantly improved the quality of $^1$H–$^{15}$N HSQC spectra yielding a central domain construct BMI1$_{121-235}$ suitable for structural studies (Fig. 1c, bottom).

Previous studies found that in cells the homology domain 1 (HD1) of PHC2 mediates the interaction with BMI1 (refs 30,40). To validate the interaction of PHC2 HD1 (residues 1–79) with the central domain of BMI1 and assess the binding affinity we employed isothermal titration calorimetry (ITC). We found that PHC2$_{1-79}$ binds to BMI1$_{121-235}$ with sub-micromolar affinity ($K_D = 398$ nM) and 1:1 stoichiometry (Fig. 1d). These results validate a direct interaction between the HD1 domain of PHC2 and the central domain of BMI1.

### BMI1 recognizes a short fragment of PHC2.

To map PHC2 residues involved in binding to BMI1 we employed NMR. We found that PHC2$_{1-79}$ is disordered in solution as judged by poor peak dispersion on the $^1$H–$^{15}$N HSQC spectra (Supplementary Fig. 3). To identify the minimal motif of PHC2 required for binding to BMI1, we again employed the carbon-detected NMR experiments. We titrated $^{13}$C,$^{15}$N PHC2$_{1-79}$ with unlabelled BMI1 and found strong broadening for a subset of resonances on 2D CACO and CBCACO spectra (Fig. 1e). This indicated that a shorter fragment of PHC2$_{1-79}$ is involved in binding to BMI1, and to identify these residues we assigned backbone chemical shifts in PHC2$_{1-79}$ using CACO, CBCACO and CANCO experiments. We found that the most significantly perturbed signals correspond to PHC2 residues 33–59 indicating that this represents the BMI1-binding motif in the PHC2 (Fig. 1f). To validate this finding we tested the binding of PHC2 fragments using ITC and found that PHC2$_{33-56}$ binds to BMI1 with a similar affinity as PHC2$_{1-79}$ ($K_D = 413$ nM) (Supplementary Fig. 4). Furthermore, deletion of residues 30–51 from full length PHC2 abolished the interaction with BMI1$_{106-326}$ and full length BMI1 in pull-down experiments performed in HEK293 cells (Fig. 1a, Supplementary Fig. 1a), further supporting that this

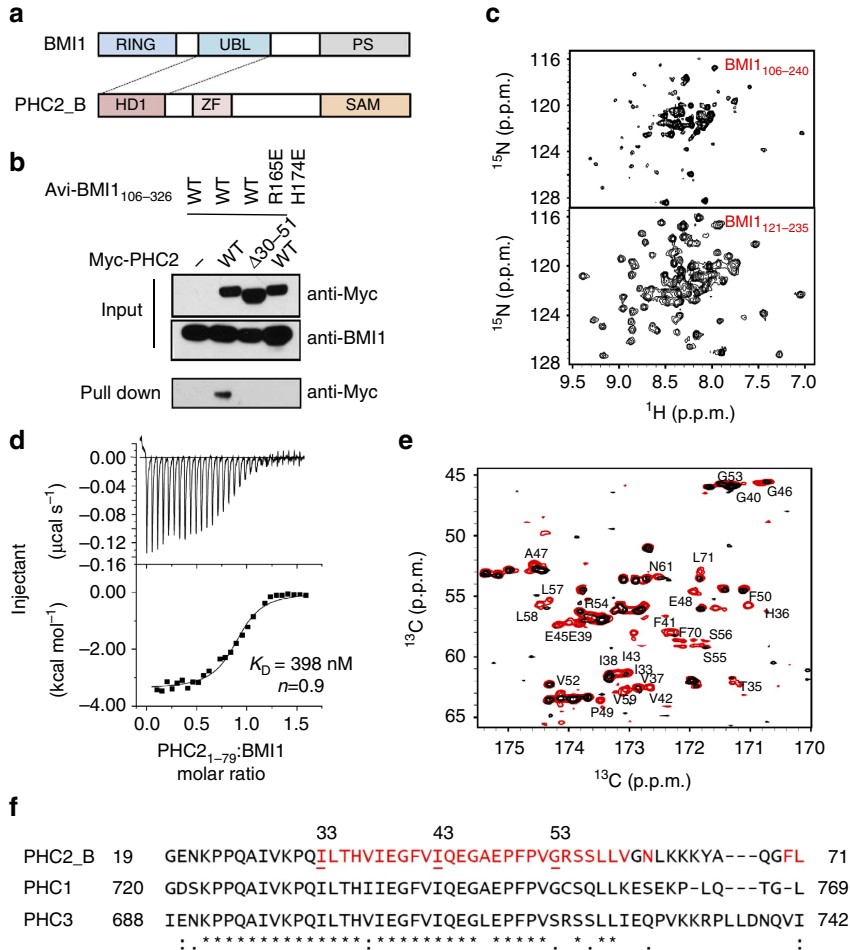

**Figure 1 | Mapping of the BMI1–PHC2 interaction.** (**a**) Schematics of the domain structures of BMI1 and PHC2. Dashed lines connect the interacting UBL and HD1 domains. (**b**) Streptavidin pull-down in HEK293 cells transfected with Avi-tagged wild-type BMI1 fragment 106–326 or R165E, H174E double mutant, BirA and Myc-tagged wild-type PHC2_B or PHC2_B with deleted residues 30–51. Western blots are probed as indicated. (**c**) Optimization of BMI1 constructs for structural studies: (top) $^1$H-$^{15}$N HSQC spectrum of BMI1$_{106–240}$; (bottom) $^1$H-$^{15}$N HSQC spectrum of BMI1$_{121–235}$. (**d**) Characterization of the affinity and stoichiometry of the BMI1–PHC2$_{1–79}$ interaction using isothermal titration calorimetry with BMI1$_{121–235}$ titrated with PHC2$_{1–79}$. (**e**) Superposition of CACO spectra for 60 μM PHC2$_{1–79}$ (red) and 60 μM PHC2$_{1–79}$ with equimolar concentration of unlabelled BMI1$_{121–235}$ (black). PHC2 residues broadened in the presence of BMI1 are labelled. (**f**) Sequence alignment of the three human PHC proteins. Residues of PHC2_B that are perturbed on addition of BMI1$_{121–235}$ are shown in red.

region is important for binding. Overall, we concluded that a relatively short, 24 amino acid fragment of PHC2 (PHC2$_{33–56}$) represents the BMI1 binding motif. This motif is strongly conserved between PHC2, PHC1 and PHC3 suggesting that BMI1 interacts with the three members of the polyhomeotic family (Fig. 1f) in a very similar manner and with similar affinities.

**Structure determination using joint X-ray and NMR refinement.** To understand the molecular basis of the BMI1–PHC2 interaction we pursued structural studies. We obtained crystals of BMI1$_{121–235}$ co-crystallized in the presence of the PHC2$_{33–56}$ fragment, which diffracted to 2.5 Å resolution (Table 1). To determine the crystal structure we used molecular replacement using a structural model derived from the BMI1 homolog PCGF1, with 31% sequence identity to BMI1 (ref. 41) (Table 1). The structure confirmed that the central domain of BMI1 adopts an ubiquitin-like (UBL) fold (Fig. 2a). While we could refine the structure of the UBL domain we were not able to model the PHC2 fragment into the remaining electron density.

The unmodelled electron density was found in a wide opening between the β2 strand and α1 helix at the interface between the two BMI1 UBL symmetry related molecules (Fig. 2a, Supplementary Fig. 5). The incomplete model of the complex is reflected by relatively high R-factor values (Table 1).

To determine the complete structure of the BMI1–PHC2 complex we turned to solution NMR. Due to limited solubility and stability of the BMI1–PHC2 complex we made a fusion protein connecting PHC2$_{30–64}$ fragment fused to the N-terminus of the BMI1 UBL domain. This construct was designed to include the intact BMI1 interacting motif in PHC2 as well as a short, 8 amino acid linker to ensure proper folding of the complex. The HSQC spectrum of the fusion protein was nearly identical to the spectrum of $^{15}$N BMI1 UBL saturated with unlabelled PHC2$_{32–61}$, demonstrating that the fusion protein recapitulates the structure of the non-covalent complex (Supplementary Fig. 6). We noted significant broadening of PHC2–BMI1 resonances at concentrations above 200 μM, which precluded complete chemical shift assignment and determination of the PHC2–BMI1 structure solely based on the NMR data. However, we obtained nearly complete backbone and methyl group

**Table 1 | Crystallographic refinement statistics for BMI1 UBL domain.**

| PDB ID | 5FR6 |
|---|---|
| *Data collection* | |
| Space group | P3$_2$12 |
| Cell dimensions | |
| a, b, c (Å) | 78.28, 78.28, 43.12 |
| α, β, γ (°) | 90.00, 90.00, 120.00 |
| Resolution (Å) | 39.41-2.50 (2.54-2.50)* |
| R$_{sym}$ | 8.7 (47.7) |
| I/σI | 33.93 (3.28) |
| Completeness (%) | 99.8 (99.6) |
| Redundancy | 10.0 (8.3) |
| | |
| *Refinement* | |
| Resolutions (Å) | 39.14-2.51 |
| No. reflections | 52,849 |
| R$_{work}$/R$_{free}$ | 23.71/32.08 |
| No. atoms | |
| Protein | 692 |
| Ligand/ion | — |
| Water | 32 |
| B-factors | |
| Protein | 57.471 |
| Ligand/ion | — |
| Water | 62.509 |
| R.m.s. deviation | |
| Bond length (Å) | 0.013 |
| Bond angle (º) | 1.783 |

BMI1, B cell-specific Moloney murine leukemia virus integration site 1; UBL, ubiquitin like.
*Values in parentheses are for highest-resolution shell.

assignment for PHC2–BMI1 and assigned a significant number of intra-PHC2 and PHC2–BMI1 NOEs. To determine the structure of the PHC2–BMI1 complex we integrated the crystal structure of the UBL domain and NMR restraints for intramolecular PHC2 and intermolecular PHC2–BMI1 contacts. The initial structure of PHC2–BMI1 was calculated in CYANA[42] followed by refinement using the Rosetta software incorporating both X-ray crystallography and NMR data[43,44]. The structure of the UBL domain was constrained during the refinement with the exception of side chains of residues 160–178 at the interface with PHC2 (Table 2). Joint refinement using the crystal structure of the BMI1 UBL domain and NMR distance restraints for PHC2 was necessary to determine the overall structure of the PHC2–BMI1 complex (Fig. 2b).

**The UBL domain binds PHC2 motif in β-hairpin conformation**. The structure of the PHC2–BMI1 complex shows that PHC2 binds into a hydrophobic site between the α1 helix and β2 strand (Fig. 2c). PHC2 residues 33–47 are well structured and adopt a β-hairpin conformation in the complex (Fig. 2c). The PHC2–BMI1 interaction involves an antiparallel β-sheet formed between the β-hairpin of PHC2 and the β2 strand of BMI1 UBL, which is stabilized by the hydrogen bonds between BMI1 Tyr163 and PHC2 Gly46. The PHC2 β-hairpin buries the hydrophobic side chains of Ile38, Phe41 and Ile43 which pack onto the hydrophobic BMI1 interface lined with residues Leu164, Cys166, Pro167, Met170, Leu175 and Phe178 (Fig. 2d). Notably, PHC2 Glu45 is buried at the interface and makes electrostatic contacts with BMI1 Arg162 and Lys182 (Fig. 2d). Mapping of PHC2 by $^{13}$C detected NMR experiments revealed that a longer fragment (PHC2$_{33–56}$) is involved in binding to BMI1. Based on the structure of PHC2–BMI1 we determined that PHC2 residues 48–56 are not well ordered in the structure, as supported by the random coil

chemical shifts and lack of long range NOEs. It is likely that these residues are involved in long range electrostatic interactions stabilizing the PHC2–BMI1 complex (for example, the interaction of PHC2 Glu48 with BMI1 Arg165).

Superposition of the PHC2–BMI1 structure with the crystal structure of BMI1 UBL shows that the PHC2 fragment overlaps with the unmodelled electron density observed in the crystal (Supplementary Fig. 7). Since PHC2 binds at the interface of two symmetry related BMI1 UBL molecules, most likely in the crystal structure one molecule of PHC2 is bound to each monomer with 50% occupancy. This unusual binding stoichiometry observed in the crystal structure precludes modelling of the complex solely based on the X-ray data.

**Point mutations in BMI1 disrupt the interaction with PHC2**. We used the structure of the PHC2–BMI1 complex to design mutations in BMI1 disrupting the interaction with PHC2. We rationalized that mutation of Arg165 and His174 in BMI1 to glutamic acids would introduce a significant electrostatic repulsion with PHC2 Glu48 and Glu39, respectively. We tested binding of these mutants to fluorescein-tagged PHC2$_{32–61}$ using a fluorescence polarization assay (Fig. 2e). While wild-type BMI1 UBL binds PHC2$_{32–61}$ with $K_D = 0.215 \pm 0.016\,\mu$M, both BMI1 mutants showed substantially reduced binding affinities. The R165E mutation reduced the binding affinity by ∼30-fold ($K_D = 5.9 \pm 0.9\,\mu$M) and the H174E mutation resulted in a 100-fold loss in the binding affinity ($K_D = 20.13 \pm 2.8\,\mu$M). Introduction of the double R165E/H174E mutation into BMI1 nearly completely abolished the interaction with PHC2 ($K_D > 50\,\mu$M). NMR analysis of the BMI1 UBL domain mutants revealed that these mutants remain folded in solution (Supplementary Fig. 8a) and confirmed that the R165E/H174E double mutant had significantly reduced binding to PHC2 (Supplementary Fig. 8b). To further validate that point mutations in BMI1 impair binding to PHC2 we performed pull-down experiments in HEK293 cells and found that the BMI1 R165E/H174E mutant does not associate with PHC2 (Fig. 1b, Supplementary Fig. 1). To further probe whether hydrophobic contacts contribute to the BMI1–PHC2 interaction we introduced M170E point mutation in BMI1 and found that it significantly reduced the binding to PHC2 by ∼80-fold ($K_D = 17 \pm 1.7\,\mu$M) (Supplementary Fig. 8c).

**BMI1 UBL forms higher order oligomers in solution**. We observed that the NMR spectra of all tested BMI1 constructs showed a concentration-dependent peak broadening consistent with protein self-association in solution. Importantly, the self-association of BMI1 was not affected by the presence of PHC2 suggesting that oligomerization may represent an intrinsic property of the BMI1 UBL domain. To characterize the oligomerization of the BMI1 UBL domain we employed analytical ultracentrifugation experiments[45]. We found a concentration dependent increase in sedimentation coefficients for the BMI1 UBL–PHC2$_{1–79}$ complex (Fig. 3a). Although we were not able to determine quantitatively the population of oligomers, these data are consistent with the propensity of the BMI1–PHC2 complex to form higher order oligomers in solution.

To investigate potential oligomerization interfaces we inspected the crystal packing of the UBL domain in the structure of the BMI1–PHC2 complex. Analysis of the crystal packing using PISA software[46] suggests two possible homo-oligomerization interfaces (Fig. 3b). The first interface is predominantly hydrophobic and comprises residues D184–F189 and Y225–T230 and has a buried surface area of 462 Å$^2$ with predicted $\Delta G = -10.4$ kcal mol$^{-1}$ for the association energy. The second interface is centered

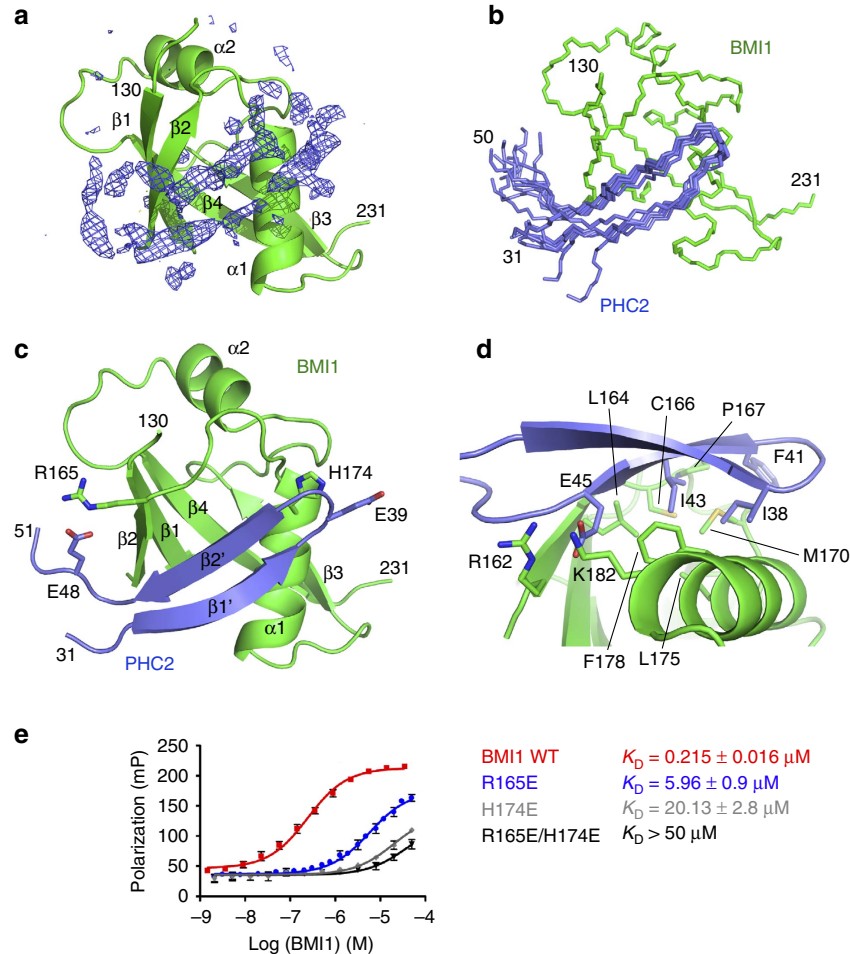

**Figure 2 | Structure of the PHC2–BMI1 complex.** (**a**) Crystal structure of the BMI1 UBL domain with Fo-Fc electron density map showing unmodelled density (blue). (**b**) Ten lowest energy structures of the PHC2–BMI1 complex determined using joint refinement employing NMR and X-ray data. The backbone of BMI1 residues 130–231 is shown in green and PHC2 residues 31–50 are shown in blue. Unstructured residues are omitted for clarity. (**c**) The overall structure of the PHC2–BMI1 complex using ribbon representation. Side chains of residues mutated to disrupt the PHC2–BMI1 interaction are shown in sticks. (**d**) Details of the PHC2–BMI1 interface. (**e**) Binding affinities of PHC2 with BMI1 variants determined using fluorescence polarization (FP) experiments titrating FITC–PHC2$_{32-61}$ with wild-type BMI1 UBL or point mutants. Experiments were performed in duplicates, error bars represent s.d.; $K_D$ is reported as an average and s.d. from three independent experiments.

around Ile212 and involves a cluster of tyrosine side chains (Tyr202, Tyr203, Tyr211 and Tyr213) that form a hydrogen bond network with backbone carbonyls. This interface is smaller, with 398 Å$^2$ buried surface area and with a predicted $\Delta G = -2.4\,\mathrm{kcal\,mol^{-1}}$ for the association energy. To probe these two potential oligomerization interfaces we designed two point mutations: F189Q and I212E with the goal to disrupt hydrophobic contacts. The $^1$H–$^{15}$N HSQC spectra demonstrate that both mutants are folded in solution (Supplementary Fig. 9a) and FP measurements indicate that these mutations do not substantially affect PHC2 binding to BMI1 ($K_D = 0.409 \pm 0.16\,\mu\mathrm{M}$ for F189Q BMI1 mutant and $K_D = 1.04 \pm 0.25\,\mu\mathrm{M}$ for I212E BMI1 mutant) (Fig. 3c). We also validated by NMR and co-immunoprecipitation that the I212E mutant binds PHC2 in a similar manner as the wild-type protein (Supplementary Figs 1 and 9a). To assess whether these point mutations affect the propensity of BMI1 UBL to homo-oligomerize we again employed analytical ultracentrifugation (Fig. 3d). Sedimentation velocity experiments showed that the BMI1 UBL F189Q–PHC2$_{1-79}$ complex has concentration dependent sedimentation coefficients similar to the wild-type protein, indicating that this mutant is still able to form higher order

oligomers. Interestingly, the sedimentation coefficients of the I212E mutant did not show such concentration dependence (Fig. 3d). This indicates that the I212E mutant has a reduced tendency for oligomerization supporting the second interface as a homo-oligomerization interface of BMI1.

We further used NMR to assess whether the I212E mutation blocks the concentration dependent broadening of NMR spectra for the BMI1–PHC2 complex. For this purpose, we titrated $^{15}$N labelled BMI1–PHC2 complex with either unlabelled wild-type or BMI1 I212E–PHC2 complex. As expected, addition of 100 μM unlabelled wild-type BMI1–PHC2 complex resulted in significant broadening of the resonances on the HSQC spectrum of 50 μM $^{15}$N BMI1–PHC2 (Fig. 3e, Supplementary Fig. 9c,d). On the contrary, addition of 100 μM unlabelled BMI1 I212E–PHC2 complex did not cause peak broadening (Fig. 3f, Supplementary Fig. 9c,d). Altogether, both NMR and analytical ultracentrifugation experiments consistently validate that mutation of Ile212 impairs homo-oligomerization of the BMI1–PHC2 complex.

**BMI1–PHC2 and BMI1–BMI1 interactions regulate PRC1 activity.** The PRC1 complex serves as an E3 ubiquitin ligase

**Table 2 | NMR restraints and refinement statistics for the PHC2–BMI1 structure.**

| PDB ID | 2NA1 |
|---|---|
| *NMR restraints* | |
| Distance constraints | |
| Total NOE | 144 |
| Intra-residue | 43 |
| Inter-residue | |
| Sequential ($|i\text{-}j| = 1$) | 45 |
| Medium-range ($|i\text{-}j| < 5$) | 11 |
| Long-range PHC2-PHC2 ($|i\text{-}j| \geq 5$) | 16 |
| Long-range PHC2-BMI1 ($|i\text{-}j| \geq 5$) | 29 |
| Hydrogen bonds | 8 |
| Total dihedral angle restraints | 26 |
| phi | 13 |
| psi | 13 |
| | |
| *Structure statistics* | |
| Violations (mean and s.d.) | |
| Distance constraints (Å) | $0.046 \pm 0.114$ |
| Dihedral angle constraints (°) | $0.070 \pm 0.800$ |
| Max. dihedral angle violation (°) | $1.83 \pm 3.86$ |
| Max. distance constraint violation (Å) | $0.88 \pm 0.09$ |
| Deviations from idealized geometry | |
| Bond lengths (Å) | 0.021 |
| Bond angles (°) | 1.8 |
| Average pairwise r.m.s.d.* (Å) | |
| Heavy | 0.5 |
| Backbone | 0.3 |
| | |
| *Ramachandran plot summary** | |
| Most favoured | 94.4% |
| Additionally allowed | 5.6% |
| Generously allowed | 0.0% |
| Disallowed | 0.0% |

BMI1, B cell-specific Moloney murine leukemia virus integration site 1; r.m.s.d., root mean square deviation; PHC2, polyhomeotic homologues.
*Pairwise r.m.s.d. was calculated among 10 refined structures for residues 33–47,127–138,161–231

for histone H2A, a mark associated with repressed gene transcription[22,23]. We sought to evaluate whether point mutations in the UBL domain that impair either the interactions with PHC2 or BMI1 homo-oligomerization affect the level of H2A ubiquitination. First, we determined the effect of BMI1 knockdown on the level of histone H2A Lys119 ubiquitination (Ub-H2A) in HeLa cells. We found reduced Ub-H2A levels on BMI1 knockdown (Fig. 4a), in agreement with the previous report[47]. We next assessed whether overexpression of either wild-type BMI1 or point mutants can rescue H2A ubiquitination observed on BMI1 knockdown. HeLa cells transfected with BMI1 small interfering RNA (siRNA) were subsequently transfected with BMI1 variants and the level of Ub-H2A was analysed by immunoblotting (Fig. 4b, Supplementary Fig. 10). Overexpression of wild-type BMI1 significantly increased Ub-H2A levels, while overexpression of the R165E/H174E or I212E mutants failed to rescue Ub-H2A levels (Fig. 4b, Supplementary Fig. 10). Interestingly, expression of the I212E mutant strongly decreases H2A ubiquitination below the level observed for the control siRNA suggesting a dominant-negative effect. The effect of the F189Q mutant was comparable to wild-type BMI1 suggesting that mutation of this potential homo-oligomerization interface involving Phe189 has no significant effects on BMI1 function (Fig. 4b, Supplementary Fig. 10). Together, these data demonstrate that point mutations in the BMI1 UBL domain which disrupt protein–protein interactions or homo-oligomerization of BMI1 impair the E3 ubiquitin ligase activity of the PRC1 complex.

**BMI1–PHC2 and BMI1–BMI1 interactions regulate BMI1.** To further test the functional importance of the BMI1 UBL domain interactions we performed clonogenic survival assays in the human osteosarcoma U2OS cell line. Both BMI1 and PHC2 are expressed at high levels in U2OS cells and have been found to associate with each other in large nuclear structures termed polycomb bodies[30,48]. First, we generated an U2OS cell line expressing inducible BMI1-targeting short hairpin RNA (shRNA; see Methods section). Induction of BMI1 shRNA expression significantly reduced BMI1 protein levels (Fig. 4c) and resulted in ∼25% reduction in colony numbers when compared to the cells expressing non-silencing shRNA (Fig. 4d, e). We then tested the effect of overexpression of either wild-type BMI1 or the point mutants (Supplementary Fig. 11). Overexpression of wild-type BMI1 significantly rescued the clonogenic potential of U2OS cells (Fig. 4f,g). On the contrary, overexpression of either R165E/H174E or I212E mutants did not rescue the clonogenic potential of U2OS cells expressing BMI1 shRNA. Expression of these two variants of BMI1 significantly decreased colony numbers when compared to wild-type BMI1, again supporting that R165E/H174E and I212E mutants function as dominant negatives (Fig. 4f,g) possibly through activity of additional PCGF members. Since the mutations in BMI1 are introduced in the UBL domain, we expect that both BMI1 variants retain the interaction with RING1A/B proteins via the N-terminal RING domain. Therefore, the strong reduction in clonogenic potential of U2OS cells results from the loss of interaction with PHC2 or from the disruption of BMI1 homo-oligomerization. Expression of the F189Q mutant did not fully rescue the clonogenic activity of BMI1 knockdown cells, suggesting that this mutation also partially impairs BMI1 function; although to a much weaker extent compared to R165E/ H174E or I212E mutants.

## Discussion

BMI1 is an essential component of the canonical polycomb repressive complex 1 (PRC1) that participates in H2A ubiquitin ligase activity[26,27] and in protein–protein interactions stabilizing the overall architecture of PRC1 (refs 29,30). In this study, we found that the central domain of BMI1 has an ubiquitin-like (UBL) fold and plays a dual role in maintaining protein–protein interactions within PRC1. First, the UBL domain interacts with the polyhomeotic protein PHC2, a member of the canonical PRC1. Second, the UBL domain is involved in homo-oligomerization of BMI1. Employing functional studies we validated that both of these activities mediated via the UBL domain are essential for BMI1 function.

Our work supports previous studies demonstrating that the UBL domain of BMI1 is involved in protein–protein interactions and is essential for the oncogenic activity of BMI1 (refs 36–38). Here, we mapped a 24 amino acid fragment of PHC2 that binds to BMI1 with a submicromolar affinity ($K_D = 398$ nM) and 1:1 stoichiometry. We determined the structure of BMI1 in complex with PHC2 and found that the UBL domain recognizes a disordered fragment of PHC2 that adopts a hairpin conformation on binding. Such architecture of the BMI1–PHC2 interaction involving the UBL domain is conserved among other protein–protein interactions of polycomb proteins, including the RING1B–CBX7, RING1B–RYBP and PCGF1–BCOR complexes[41,49].

Sequence alignment of BMI1 homologues (PCGF family) shows that the UBL domain of BMI1 is most similar to MEL18 (PCGF2) with 60% sequence identity, while it differs significantly from the remaining homologues (Supplementary Fig. 12a,b). To assess whether the MEL18 UBL domain interacts with PHC2 we tested the binding of MEL18$_{121\text{-}237}$ fragment with PHC2$_{32\text{-}61}$

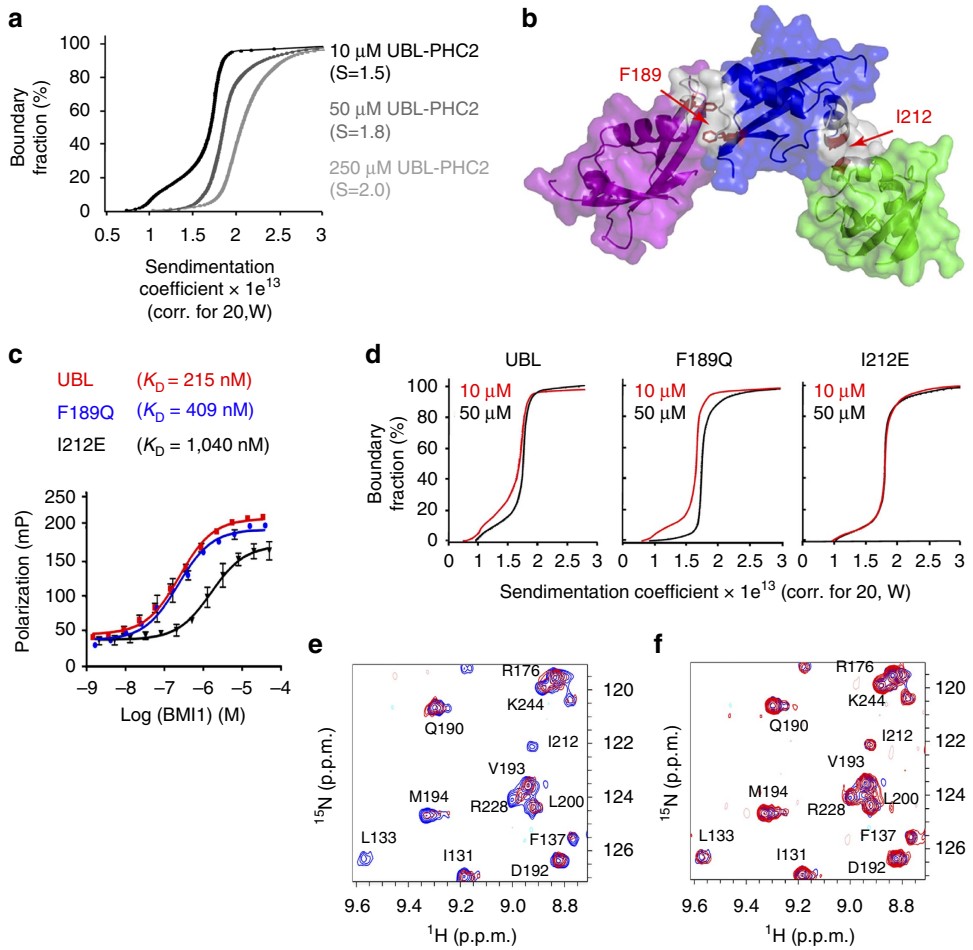

**Figure 3 | BMI1 UBL forms higher order oligomers in solution.** (**a**) Van Holde-Weischet [G(s)] plots of sedimentation distributions for wild-type BMI1 UBL–PHC2$_{1-79}$ complex showing a concentration dependent increase in particle size in solution. Values of sedimentation coefficients are in parenthesis. (**b**) The crystal structure of the BMI1 UBL domain showing two putative homo-oligomerization interfaces identified based on crystal packing. Residues selected for mutation to block homo-oligomerization are labelled. (**c**) Binding affinities of BMI1 variants with PHC2$_{1-79}$ determined using fluorescence polarization experiments titrating FITC–PHC2$_{32-61}$ with wild-type BMI1 UBL or point mutants. Experiments were performed in duplicates, error bars represent s.d.; $K_D$ is reported as the average and s.d. from three independent experiments. (**d**) Van Holde-Weischet [G(s)] plots of sedimentation distributions for BMI1 UBL–PHC2$_{1-79}$ complexes at 10 and 50 μM comparing the wild-type BMI1 and point mutants F189Q and I212E. (**e**) Superposition of $^1$H-$^{15}$N HSQC spectra for 50 μM $^{15}$N BMI1 UBL–PHC2$_{33-56}$ complex in the absence (blue) and presence (red) of 100 μM unlabelled BMI1 UBL–PHC2$_{33-56}$ complex (red). (**f**) Superposition of $^1$H-$^{15}$N HSQC spectra for 50 μM $^{15}$N BMI1 UBL–PHC2$_{33-56}$ complex in the absence (blue) and presence of 100 μM unlabelled BMI1 UBL I212E–PHC2$_{33-56}$ complex (red).

using the FP assay. We found that MEL18 binds PHC2 with slightly reduced affinity ($K_D = 812 \pm 9.0$ nM) when compared to BMI1 (Supplementary Fig. 12c). Analysis of the BMI1–PHC2 structure revealed that all BMI1 residues involved in contacts with PHC2 are identical with MEL18 and strongly suggests very similar binding mode for MEL18–PHC2. Sequences of the remaining homologues of BMI1 differ significantly and most likely they do not interact with PHC proteins.

Our studies identified that BMI1 has a propensity to form homo-oligomers in solution. We employed analytical ultracentrifugation to demonstrate that BMI1 UBL domain undergoes concentration dependent oligomerization. Analysis of the crystal packing for the BMI1–PHC2 structure suggested two possible homo-oligomerization interfaces, one of which harboring Ile212 was validated through mutagenesis studies. Interestingly, our finding is consistent with the previous observations that full length RING1B and BMI1 form a heterotetramer *in vitro*[26].

Structure determination of the BMI1–PHC2 complex represented a significant technical challenge. Previous attempts at structural studies of the BMI1 UBL domain were hampered by

protein instability[41]. To design constructs of BMI1 and PHC2 suitable for structural studies we employed $^{13}$C-detected NMR experiments to identify disordered protein fragments[39]. Using such an approach we efficiently mapped the minimal folded fragments of BMI1 and PHC2 involved in this interaction highlighting the utility of this methodology in designing constructs for structural biology[39]. Furthermore, structure determination of the BMI1–PHC2 complex solely by either X-ray crystallography or solution NMR was hampered by additional challenges, such as crystallographic artifacts from the crystal packing, low protein solubility and high propensity for aggregation. To resolve these difficulties we used a Rosetta-based joint refinement method incorporating the crystal structure of the BMI1 UBL domain and NMR restraints defining UBL–PHC2 contacts. While joint refinement using both methods has been used before to improve structure accuracy[50–53]; to our knowledge, our work represents a unique example where X-ray crystallography data for a protein–protein complex were not sufficient for structure determination and NMR restraints were required to determine the overall structure of the complex.

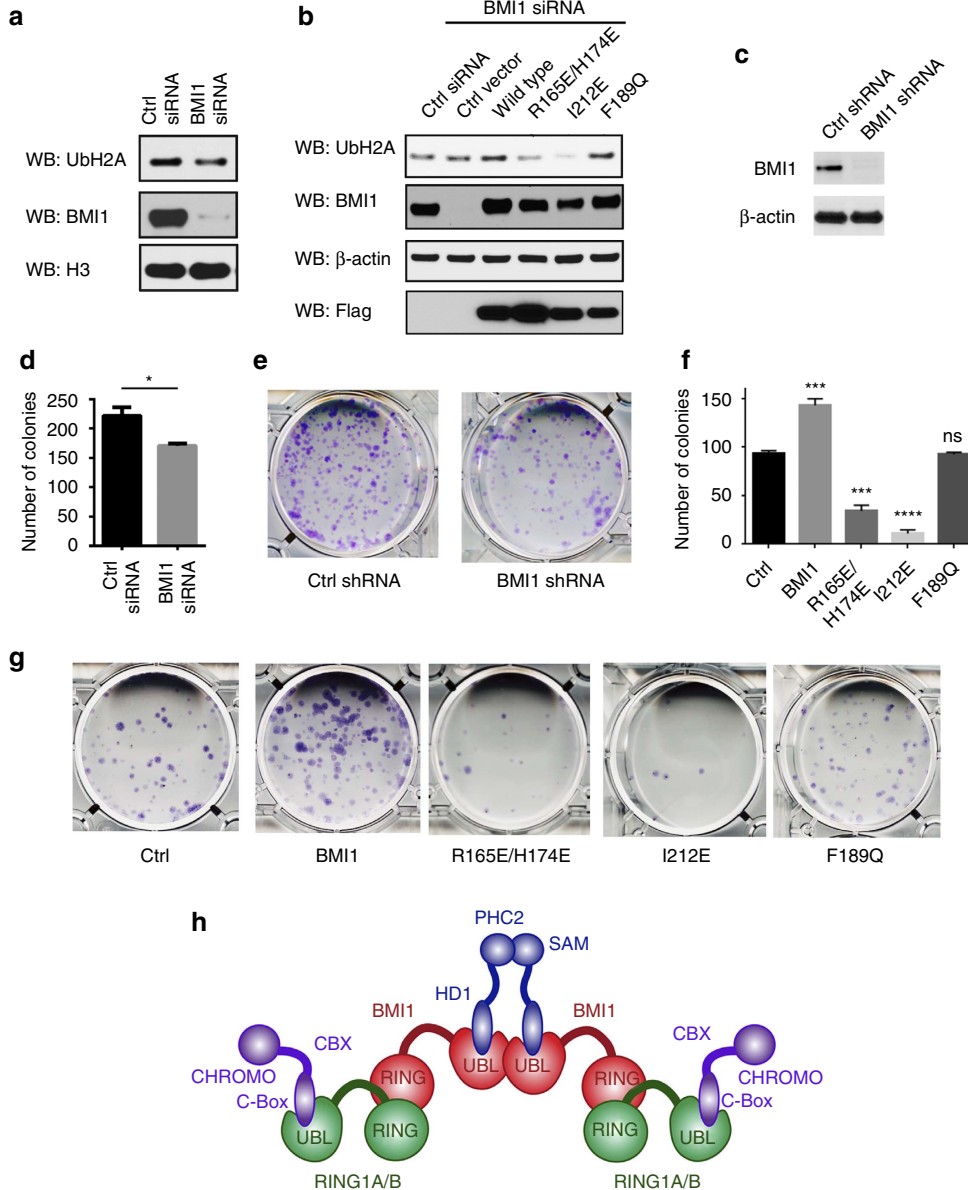

**Figure 4 | Functional consequences of disrupting BMI1 protein–protein interactions and homo-oligomerization.** (**a**) Analysis of the effect of BMI1 knockdown on Ub-H2A levels in HeLa cells. HeLa cells were transfected with control or BMI1 3′ UTR siRNA and followed by immunoblot analysis 48 h post transfection. (**b**) Characterization of Ub-H2A levels on overexpression of BMI1 mutants in HeLa cells. HeLa cells transfected with control or BMI1 3′ UTR siRNA for 48 h were transfected with plasmids encoding Flag-tagged full length wild-type BMI1 or three mutants and analysed using immunoblotting after 48 h. (**c**) Inducible knockdown of BMI1 in U2OS cells using shRNA. U2OS cells carrying control or BMI1 shRNA were treated with doxycycline for 96 h and cell lysates were analysed for BMI1 expression levels. (**d**) Quantification of colony numbers from the clonogenic survival assay in panel **e** demonstrating the effect of BMI1 knockdown on clonogenic potential of U2OS cells. U2OS cells were plated in triplicates and incubated for 14 days. Error bars represent s.d. of triplicate wells; *$P \leq 0.05$. (**e**) Representative plates from clonogenic survival assays on BMI1 knockdown in U2OS cells. (**f**) Quantification of the clonogenic survival assay in panel **g** demonstrating the effect of re-expression of BMI1 mutants in U2OS cells expressing BMI1 shRNA. Error bars represent s.d. of triplicate wells; ***$P \leq 0.001$; ****$P \leq 0.0001$; ns, not significant. (**g**) Representative plates from the clonogenic survival assay on BMI1 knockdown and re-expression of BMI1 mutants in U2OS cells. (**h**) Proposed architecture of the PRC1 complex.

On the basis of our structural studies, we generated and characterized point mutations in the BMI1 UBL domain, which disrupt the interaction with polyhomeotic proteins as well as BMI1 UBL homo-oligomerization. The BMI1 R165E/H174E double mutant has significantly reduced binding to PHC2 and the I212E mutant has strongly decreased propensity to homo-oligomerize. We found that both mutants impair the activity of PRC1 to ubiquitinate H2A in HeLa cells. Furthermore, knockdown of BMI1 followed by re-expression of BMI1 R165E/H174E or I212E mutants strongly decreases the clonogenic

potential of U2OS cells. These findings emphasize that both the protein–protein interactions and oligomerization mediated through the UBL domain are necessary for BMI1 function. The point mutations we characterized provide a potential tool which might be further explored to dissect functional implications of disrupting the BMI1 interactions.

Our structural studies established an important role of BMI1 in coordinating the architecture of the canonical PRC1 complex. BMI1 is a key structural component of the PRC1 that simultaneously interacts with RING1A/B and PHC proteins

bridging the H2A ubiquitin ligase activity of RING1A/B and the polyhomeotic subunit (Fig. 4h). The interaction of the BMI1 UBL domain with PHC proteins is essential for the proper assembly of the PRC1 complex which is needed for ubiquitination of H2A in cells. To further validate this, we have performed knockdown of PHC2 and PHC3 and found a very significant loss of H2A ubiquitination in HeLa cells on simultaneous loss of PHC proteins (Supplementary Fig. 13). We also found that the BMI1 UBL domain may represent a motif contributing to the oligomerization of PRC1. While BMI1 UBL homo-oligomerization is relatively weak *in vitro*, the UBL domain may function in concert with SAM domains, a well-known oligomerization motif in the PHC subunit (Fig. 4h)[54], which is likely involved in PRC1 clustering[40]. As previously suggested, PRC1 may regulate gene expression by compacting chromatin through interactions between distinct PRC1 complexes at either local or distal nucleosomes[24,40]. Here we propose a model where the homo-oligomerization properties of BMI1 enhance the higher order architecture of PRC1 (Fig. 4h).

Emerging studies suggest that BMI1 is a promising target for the development of anti-cancer therapeutics[16]. Detailed characterization of BMI1 protein–protein interactions will support the design of efficient strategy to reverse the oncogenic activity of BMI1. The structure of the PHC2–BMI1 complex we determined here provides the molecular basis to develop small molecule inhibitors of BMI1 activity and might pave the way towards novel anti-cancer therapeutics.

## Methods

**Plasmid construction.** cDNA encoding BMI1$_{106-240}$, BMI1$_{121-235}$ and MEL18$_{121-237}$ (ordered from Life Technologies) and full-length PHC2_B (a kind gift from Dr Jeff Rual, UM Pathology) were subcloned into a modified pET32a vector with a N-terminal His$_6$-thioredoxin expression tag and PreScission protease cleavage site. The cDNA encoding the PHC2$_{30-64}$–BMI1$_{121-235}$ fusion was ordered from Life Technologies and subcloned into the same vector. The mammalian expression vectors for pCMV BMI1 and PHC2 constructs were prepared from cDNA using standard subcloning techniques. The genes for the *E. Coli* BirA biotin ligase and Avi–BMI1$_{106-326}$ were ordered from Genscript and Life Technologies, respectively, and subcloned into the pCMV vector. Mutant constructs were generated using QuikChange site-directed mutagenesis protocol (Agilent).

**Protein expression and purification.** BMI1 UBL domain variants, MEL18 UBL domain and PHC2–BMI1 fusion were expressed in Codon+ BL21 (DE3) *E. coli* cells (Sigma) with an N-terminal His$_6$- thioredoxin tag. Cells were grown in Luria broth or labelled M9 medium with ampicillin selection. After 16 h induction with 0.5 mM IPTG at 18 °C cells were resuspended in lysis buffer (50 mM tris, pH 7.5 at 25 °C, 250 mM NaCl, 20 mM Imidazole, 0.5 mM phenylmethylsulfonyl fluoride (PMSF), 1 mM β-mercaptoethanol (β-ME) and lysed using a cell disrupter. Clarified lysate was applied to a HisTrap HP (GE Healthcare) and eluted with lysis buffer containing 0.5 M imidazole. To remove the His$_6$- thioredoxin tag, the protein was cleaved with PreScission protease and BMI1 constructs were further purified using cation exchange chromatography. Purified protein was buffer exchanged into storage buffer (100 mM bis tris, pH 6.5, 50 mM NaCl, 1 mM TCEP) using HiPrep Desalting Column (GE Healthcare). For crystallization, BMI1 was incubated with two-fold excess of a synthetic PHC2$_{33-56}$ peptide (Genscript) and applied to a Superdex S75 gel filtration column (GE Healthcare) pre-equilibrated with storage buffer.

PHC2$_{1-79}$ was expressed in BL21 (DE3) *E. coli* cells (Sigma) with an N-terminal His$_6$- thioredoxin tag. Cells were grown in Luria broth or labelled M9 medium with ampicillin selection. After 16 h induction with 0.5 mM IPTG at 18 °C cells were resuspended in lysis buffer (50 mM tris, pH 7.5 at 25 °C, 250 mM NaCl, 20 mM imidazole, 0.5 mM phenylmethylsulfonyl fluoride (PMSF), 1 mM β-mercaptoethanol (β-ME)) and lysed using a cell disrupter. Clarified lysate was applied to a HisTrap HP (GE Healthcare) and eluted with lysis buffer containing 0.5 M imidazole. The His$_6$- thioredoxin tag was cleaved with PreScission protease during dialysis against 100-fold excess dialysis buffer (50 mM tris, 50 mM NaCl, 1 mM TCEP at pH 7.5) and separated from PHC2$_{1-79}$ by nickel affinity. PHC2$_{1-79}$ was further purified by Superdex S75 gel filtration (GE Healthcare) pre-equilibrated with storage buffer (50 mM phosphate, pH 6.5, 50 mM NaCl, 1 mM TCEP).

**Isothermal titration calorimetry.** The measurements were performed using a VP-ITC titration calorimetric system (MicroCal) at 25 °C. BMI1$_{121-235}$ and trx-PHC2$_{1-79}$ were dialysed extensively against ITC buffer (50 mM phosphate, pH 6.5, 50 mM NaCl, 1 mM TCEP). Peptide PHC2$_{33-56}$ (Genscript) was directly dissolved in ITC buffer. All samples were degassed before measurements. For

measurement of BMI1$_{121-235}$–PHC2$_{1-79}$ interaction the calorimetric cell, containing BMI1 (22 μM), was titrated with trx-PHC2$_{1-79}$ (164 μM) injected in 10 μl aliquots. For measurement of BMI1$_{121-235}$–PHC2$_{33-56}$ interaction the calorimetric cell, containing BMI1 (9 μM), was titrated with PHC2$_{33-56}$ (100 μM) injected in 10 μl aliquots. Data were analysed using Origin 7.0 (OriginLab) to obtain $K_D$ and stoichiometry.

**Protein NMR experiments.** Samples for assignment of $^{13}C^{15}N$–BMI1$_{106-240}$ contained 70 μM protein in NMR buffer (50 mM bis tris, pH 6.5, 50 mM NaCl, 1 mM TCEP and 5% D$_2$O). 2D $^{13}C$-detected CACO, CBCACO and CANCO experiments were used for carbon assignment of flexible residues[55]. Samples for assignment of PHC2$_{1-79}$ contained 60 μM $^{13}C^{15}N$–PHC2$_{1-79}$ in a buffer containing 100 mM bis tris, pH 6.5, 100 mM NaCl, 1 mM TCEP and 5% D$_2$O. 2D $^{13}C$-detected CACO, CBCACO and CANCO experiments were used for carbon assignment[55]. Mapping the PHC2 motif involved in BMI1 binding was performed for 60 μM $^{13}C^{15}N$–PHC2$_{1-79}$ mixed with 60 μM BMI1 UBL. BMI1 UBL–PHC2 binding experiments contained 50–100 μM $^{15}N$-labelled wild-type BMI1$_{121-235}$ or mutants and unlabelled PHC2$_{33-56}$. Analysis of peak broadening and aggregation was performed by titration of 50 μM $^{15}N$-labelled BMI1$_{121-235}$ in complex with PHC2$_{33-56}$ with 50 and 100 μM unlabelled BMI1$_{121-235}$ wild-type or I212E–PHC2$_{33-56}$ complex. All NMR spectra were acquired at 303 K on a 600 MHz Bruker Advance III spectrometer equipped with cryoprobe, running Topspin version 2.1. Spectra processing and visualization were performed using NMRPipe[56] and Sparky[57].

**Crystallization and structure determination.** Initial crystals were obtained through sitting drop screening of BMI1$_{121-235}$–PHC2$_{33-56}$ complex purified using size exclusion. Crystals were further optimized by hanging-drop vapor diffusion with equal volumes (1 μl) of protein (9 mg/ml in 50 mM bis tris, pH 6.5, 50 mM NaCl, 1 mM TCEP) and the precipitant solution (100 mM MES, pH 6.5, 50 mM MgCl$_2$, 7% isopropanol, 6% PEG 4000). Crystals formed within 7 days at 4 °C. Crystals were cryoprotected using the precipitant solution containing 20% glycerol and flash frozen in liquid nitrogen. X-ray diffraction data of crystals were collected at a resolution of 2.5 Å at the Advanced Photon Source at LS-CAT beam line 21-ID-F. The data were indexed, integrated, and scaled using the HKL2000 suite[58]. The structure was determined by molecular replacement method with the CCP4 version of MOLREP[59] using the polycomb group Ring finger protein complex structure (PDB code 4HPM B chain) as a search model[41]. Model building was performed manually using the program WinCoot[60] and the refinement was performed with CCP4 refmac5 (ref. 61). The data statistics are summarized in Table 1.

**Structure determination using joint NMR and X-ray refinement.** NMR experiments for structure determination were collected for 200 μM $^{13}C^{15}N$–PHC2$_{30-64}$–BMI1$_{121-235}$ fusion protein in 50 mM bis tris, pH 6.5 buffer with 50 mM NaCl, 1 mM TCEP and 5% D$_2$O. Backbone assignment was completed based on a series of triple-resonance experiments including HNCACB, CBCA (CO)NH, HNCA, HN(CO)CA, HNCO and HN(CA)CO. Methyl side chain resonances were assigned using 3D $^{13}C$-$^1H$-$^1H$ HCCH-TOCSY. Distance restraints were obtained from 3D $^{15}N$-separated NOESY-HSQC and 3D $^{13}C$-separated NOESY–HSQC spectra measured with 150 ms mixing time. Initial structures were calculated in CYANA[42] based on distance restraints from NOESY spectra and dihedral angle restraints from TALOS + (ref. 62). NMR structures of PHC2 were combined with the X-ray crystal structure of BMI1 UBL domain and the entire complex was refined using Rosetta[43,44,63] constrained by NOE-derived distances restraints, dihedral angle restraints and BMI1 crystal structure coordinates. BMI1 was restricted during refinement with the exception of loop residues 121–127 and 138–160 which were missing in the crystal structure. These fragments were added using the loop building protocol[64] and were treated as disordered fragments during Rosetta refinement. Side chains of BMI1 residues 160–178 were unrestricted during the refinement. Data statistics are summarized in Table 2.

**Fluorescence polarization assays.** Dissociation constants for binding of PHC2 to BMI1 UBL domain variants and MEL18 UBL were determined by fluorescence polarization. Fluorescein-labelled PHC2$_{32-61}$ (Genscript) at 20 nM was titrated with a range of BMI1 or MEL18 concentrations in the FP buffer (50 mM bis tris, pH 7.5, 50 mM NaCl, 1 mM TCEP, 0.01% BSA, 0.25% tween-20). After 1 h incubation of the protein–peptide complexes, changes in fluorescence polarization and anisotropy were measured at 525 nm after excitation at 495 nm using PHERAstar microplate reader (BMG). Results were used to calculate binding affinity ($K_D$) for PHC2 with wild-type BMI1, BMI1 mutants, or MEL18 using the Prism 4.0 (GraphPad) program.

**Analytical ultracentrifugation.** Sedimentation velocity experiments were performed on a Beckman Optima XL-I at the Center for Analytical Ultracentrifugation of Macromolecular Assemblies (CAUMA) at the University of Texas Health Center at San Antonio. Calculations were performed with the UltraScan software[65] at the Texas Advanced Computing Center at the University of Texas at Austin and at the Bioinformatics Core Facility at the University of Texas

Health Science Center at San Antonio. Protein samples were prepared by mixing BMI1 UBL with PHC2$_{1-79}$ followed by size exclusion to purify 1:1 stoichiometric complex. Samples were subsequently concentrated to 10, 50 or 250 µM in a 50 mM phosphate buffer, pH 6.5 containing 50 mM NaCl and 1 mM TCEP. All analytical ultracentrifugation data were collected at 20 °C and spun at 50 k r.p.m., using standard Epon-2 channel centerpieces. All data were first analysed by two-dimensional spectrum analysis with simultaneous removal of time-invariant noise[66] and then by enhanced van Holde-Weischet analysis[67] and genetic algorithm refinement[68] where applicable, followed by Monte Carlo analysis[69].

**Cell cultures.** Human embryonic kidney-293 (HEK293) (CRL-1573) cell line, cervical carcinoma cell line (HeLa) (CCL-2) and osteosarcoma cell line U2OS (HTB-96) were obtained from the American Type Culture Collection and were cultured either in Dulbecco's modified Eagle's medium (for HEK293 and HeLa) or McCoy's 5a Medium Modified (for U2OS) supplemented with 10% FBS. For plasmid transfection, Lipofectamine 2000 (Invitrogen) and, for the siRNA transfection, Lipofectamine RNAi-max (Invitrogen) were used according to the manufacturer's instructions.

**Pull-down and co-immunoprecipitation experiments.** HEK293 cells were transfected with BirA, Myc-PHC2_B and Avi-BMI1$_{106-326}$ constructs using Fugene 6 (Roche) transfection agent. Cells were harvested by centrifugation 48 h after transfection and lysed through sonication in lysis buffer (50 mM HEPES, pH 7.5, 150 mM NaCl, 1 mM EDTA, 2.5 mM EGTA, 1 mM NEM, 1 mM NaF, 0.1 M Na$_3$VO$_4$, 10% glycerol, 0.1 mM β-glycerophospate, 0.01% NP-0.4) with protease inhibitor cocktail (Sigma). Lysate was clarified by centrifugation and streptavidin magnetic beads (Pierce) were added to each sample and incubated at 4 °C with rotation for 16 h. Beads were washed 4 times with wash buffer (20 mM tris, pH 8.0, 300 mM KCl, 1 mM EDTA, 10% glycerol, 0.1% NP-0.4) with protease inhibitor cocktail (Sigma) and proteins were boiled in Lammeli buffer. Samples were analysed by SDS–PAGE and western blotting probed with either Myc antibody (Cell Signaling, catalog #2276S) or BMI1 antibody (Millipore, catalog #05-637). 10% of the total protein used for pulldown was taken as input control. Uncropped images of the western blots are shown in Supplementary Fig. 14.

For co-immunoprecipitation experiments HEK293 cells were transfected with full length Flag-tagged BMI1, R165E/H174E, I212E mutants and Myc-PHC2_B constructs using lipofectamine 2000 (Invitrogen). After 48 h incubation, cells were harvested and lysed and processed as described above. Flag-M2 dynabeads (Sigma) were washed in lysis buffer and added to each sample and incubated at 4 °C with rotation overnight. Beads were washed 3 times with lysis buffer and boiled in 1% SDS. Samples were mixed with Lammeli buffer, boiled and were analysed by SDS–PAGE and western blotting with Myc (Cell Signaling), Flag (Sigma, catalog #F3165), BMI1 (Millipore) and β-Actin antibodies. 10% of the total protein used for immunoprecipitation was used as an input control.

**Lentiviral shRNA-mediated gene knockdown.** Control and BMI1 shRNA expressing U2OS stable transfectant cells were generated using Inducible TRIPZ Lentiviral shRNA system from Dharmacon. Briefly, various individual clones targeting different regions of BMI1 (Clone ID; V2THS_48576, V2THS_244779, V3THS_400015, V3THS_302126) and non-silencing control (shRNA) lentiviral constructs were obtained, packaged and lentiviral particles were produced as per manufacturer's instructions. U2OS cells transduced with individual constructs were continuously cultured in puromycin to select for cells containing the constructs. Stable U2OS cells were incubated in Doxycycline (1 µg ml$^{-1}$) for 48 h before assessment for % RFP expression to confirm presence of lentiviral DNA. Based on knockdown efficiency of BMI1, as determined by RFP expression and quantitative real time PCR, U2OS cells carrying BMI1 V3THS_400015 clone was selected for further analysis.

**BMI1 knockdown and analysis of ubH2A in HeLa cells.** 100,000 HeLa cells were reverse transfected with 25 nM of BMI1 3′UTR siRNA (Dharmacon) for 48 h using lipofectamine RNAiMax (Invitrogen) as per manufacturer instructions in 6 well plate. After 48 h of siRNA transfection, medium was replaced and BMI1 wild-type or mutant constructs were transfected using lipofectamine 2000 (Invitrogen) as per manufacturer protocol. Cells were incubated for 48 h before washing, trypsinization and lysis. Equal amounts of whole cell lysate were separated on 10% SDS–PAGE gels. Blots were probed with primary antibodies (H2Aub(K119), Cell Signaling Technology, catalog #8240S; BMI1, Millipore, catalog #04-1116; H3, Abcam, catalog #ab1791) overnight at 4 °C, washed five times in TBS plus 0.1% Tween (TBST) and then incubated with the appropriate horseradish peroxidase-conjugated secondary antibody for 1 h at room temperature. Membranes were washed five times in TBST and visualized on autoradiography film after incubating with ECL reagent (ECL Prime, GE Healthcare). Uncropped images of the western blots are shown in Supplementary Fig. 14.

**siRNA mediated PHC2/PHC3 knockdown in HeLa cells.** For PHC2 and PHC3 knockdown 100,000 HeLa cells were reverse transfected with either 50 nM of PHC2

and PHC3 siRNAs individually (Dharmacon) or in combination for 96 h using Lipofectamine RNAiMax as per manufacturer instructions. Equal amounts of whole cell lysate were separated on 10% SDS–PAGE gels. Blots were probed with primary antibodies (H2Aub(K119), Cell Signaling Technology; H3 (Abcam); β-actin, EMD Millipore). PHC2 and PHC3 siRNA mediated knockdown efficiency was measured by quantitative real time PCR. Uncropped images of the western blots are shown in Supplementary Fig. 14.

**Clonogenic survival assay.** Clonogenic survival assays were performed using techniques described previously[70]. Briefly, U2OS cells containing control or BMI1 shRNA, were incubated with Doxycycline (1 mg ml$^{-1}$) for 48 h to induce shRNA expression. After 48 h incubation, cells were transfected with BMI1 wild-type or mutant DNA using lipofectamine 2000 (Invitrogen) as per manufacturer protocol for another 48 h. On completion of 48 h with DNA transfection, cells were washed with PBS twice, trypsinized, counted and plated at low density (500 cells each well) in triplicates in 6 well plates and left to grow for 14 days at 37 °C. On the 14th day plates were fixed with methanol-acetic acid, stained with crystal violet, and scored for colonies containing more than 50 cells to assess the colony-forming ability. Error bars are from triplicate samples. Differences between control and BMI1 wild-type or mutant samples were analysed by unpaired $t$-test and $P$ values were estimated based on this model. Significance was determined based on a significance level of 0.05. Remaining cells were used for whole cell lysate isolation and immunoblotting analysis to determine extent of BMI1 knockdown and to confirm overexpression of BMI1 DNA constructs.

**Data availability.** The atomic coordinates and structure factors for the BMI1 UBL domain and the PHC2–BMI1 complex have been deposited in the PDB with accession codes: 5FR6 and 2NA1. The data that support the findings of this study are available within the article and its Supplementary Information files, or available from the corresponding author on request.

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

## Acknowledgements

This work was funded by the National Institute of Health (NIH) grants R01 CA181185 to T.C., R01 CA160467 and R01 CA201204 to J.G. and R01 CA134604 to E.R.L.; American Cancer Society Research Scholar Grants RSG-11-082-01-DMC to T.C. and RSG-13-130-01-CDD to J.G., and Leukemia and Lymphoma Society Scholar grants (1340-17) to T.C. and (1215-14) to J.G. Authors thank Dr Rual (Department of Pathology, University of Michigan) for PHC2_B plasmid. Use of the Advanced Photon Source was supported by the US Department of Energy, Office of Science, Office of Basic Energy Sciences under contract number DE-AC02-06CH11357. Use of the LS-CAT Sector 21 was supported by the Michigan Economic Development Corporation and the Michigan Technology Tri-Corridor for the support of this research program (grant 085P1000817).

## Author contributions

F.G. purified proteins, performed and analysed biochemical, NMR and X-ray crystallography experiments and determined the structure; H.J.C. performed X-ray crystallography experiments and determined the crystal structure, S.S. performed biology

experiments; A.H. generated U2OS cells with BMI1 shRNA; B.B. purified proteins; Ł.J. and M.J. performed and analysed NMR experiments; B.D. performed analytical ultracentrifugation experiments; F.G., E.R.L., J.G. and T.C. planned the experiments and wrote the manuscript with an input from all authors.

## Additional information

**Competing financial interests:** Drs Grembecka and Cierpicki receive research support from Kura Oncology. They are also receiving compensation as members of the scientific advisory board of Kura Oncology, and they have an equity ownership in the company. Other co-authors declare no potential conflict of interest.

