## [Peer Review File · Nature Communications]

Reviewers' comments:

Reviewer #1 (Remarks to the Author):

The manuscript titled "BMI1 regulates PRC1 architecture and activity through homo- and hetero-oligomerization" by Gray et. al. investigates the structure of complex between BMI1-UBL domain and HD1 region of PHC2. The UBL domain adopts an expected ubiquitin-like structure, it binds a short 24 residue beta-hairpin of PHC2-HD1. The authors have found that the Bmi-UBL domain can also homo-oligomerize. The hetero- and homo-oligomerization are important for in H2A ubiquitination and clonogenic activity. This is a nice structure-function study. However, there are two major concerns regarding the paper. I have provided them below along with the minor issues.

Major Issues

1. The authors have claimed "point mutations in the BMI1 UBL domain which disrupt protein-protein interactions or homo-oligomerization of BMI1 impair the E3 ubiquitin ligase activity of the PRC1 complex". (Page 11, line 12)

Comments: It is clear that the hetero- and homo-oligomerization is important for PRC1's ability to ubiquitylate H2A. However, the paper does not provide a mechanistic understanding of the role of these interactions towards H2A ubiquitylation. Does it impact the catalytic activity, E2/E3 interaction, E3/substrate (H2A) interaction or something else? There are no experiments to implicate any of these.

2. The authors have claimed in discussion (Page 13 21) that "to our knowledge, our work represents a first example that requires a combination of X-ray crystallography and NMR data to determine the overall structure of a protein-protein complex". In their Materials and Methods the structure determination part states "NMR structures of PHC2 were combined with the X-ray crystal structure of BMI1 UBL domain and the entire complex was refined using Rosetta 45,46,67 constrained by NOE-derived distances restraints, dihedral angle restraints and BMI1 crystal structure coordinates"

Comments: These calculations by taking the NMR structure of one protein, x-ray structure of the other and forming the complex using intermolecular restraints is routinely done (Pruneda et. al., Mol cell, 2012). In fact, algorithms like HADDOCK are famous for that. The author's claim of novelty is not correct.

Minor Issues

1. Page 8, line 5. The authors have stated that "likely these residues are involved in long range interactions stabilizing the PHC2-BMI1 complex."

Comments: The chemical shift perturbations observed in 48-56 could be due to minor structural changes in that region caused by the impact of the binding of the 33-47 part. How do the authors speculate that this is due to long-range interactions?

2. In the designs of mutations to disrupt BMI1:PHC2 interface, only electrostatic interactions were targeted. Are the hydrophobic interactions dispensable? Page 10, line 1. The Kd for I212E mutant does not match with the Figure 3C.

3. Page 10, line 1. The Kd for I212E mutant does not match with the Figure 3C.

4. Page 10 line 6. The difference in line-broadening is not evident from the 2D plots. Please provide cross-sections.

5. The authors have not presented statistics and validation of Rossetta generated structures.

Reviewer #2 (Remarks to the Author):

The manuscript by Gray and colleagues provides a structural model for PCGF4/BMI1 interaction with HD1 domain in Polyhomeotic paralogs. It also describes, structurally the interaction between PCGF4/BMI1 surfaces responsible for a putative oligomerization.

The data are of interest to all researchers interested in Polycomb function and contributes to the disentanglement of biochemical complexity(es) of type I Polycomb Repressive Complexes.

I believe, however, that studies on the functional implications inferred from new structures should be reinforced/clarified.

For instance, binding of truncated BMI1 protein to PHC2 in Fig. 1B demonstrates that the deleted region (1-105 amino acids deleted) is dispensable for such an interaction. It would have been informative, however, to compare it with binding to full length BMI1. I'd guess a larger amount of PHC2 would be co-immunoprecipitated, considering the apparent low (in a transiently transfected 293 cell system) amount of PHC2 bound to truncated BMI1.

Experiments in Fig 4: it can be understood that wild type and mutant BMI1 cDNAs are full length forms but it doesn't harm to clarify this. Also, it would have been convenient that cDNAs were tagged so that they could be distinguished from endogenous BMI1 protein. I would imagine that siRNA treatment (Fig. 4B) may have a transient effect on the endogenous protein and, if as I understand, plasmid transfection is in the absence of siRNA re-expression of endogenous BMI1 would be occurring.

Although the differences in UbH2A are clear it would have been convenient that transfection efficiencies could be normalised. Also, importantly, western blot measuring global UbH2A levels corresponding to longer exposures will allow a better evaluation of the mutations investigated. A concern is that although decreased, UbH2A levels remain functionally significant. That BMI1 variants have an impact on clonogenicity (no data for proliferation is given, despite the text says that proliferation is affected) may not necessarily be determined only by alteration, particularly if modest, of UbH2A levels. Unless powerful dominant effects are taking place, it has to be considered the presence of additional PCGF members acting as positive cofactors of RING1 E3 ligase.

It would also have been desirable to discuss about the specificity of interactions of HD1 domains in PHC paralogs with PCGF2 (not studied here) and PCGF4. Surely, if PHC proteins bind BMI1 sequences present in all PCGF paralogs there is got to be other regions in these PCGF proteins that help discriminate interactions.

Interestingly, a mutation in the BMI1 oligomerization region has large impacts in UbH2A and clonogenicity assays: would it affect binding to PHC2?

We would like to thank Reviewers for their very positive comments. We have performed new experiments to address all the points raised by the Reviewers to further strengthen the manuscript. Specific comments are below.

Reviewers' comments:

Reviewer #1 (Remarks to the Author):

The manuscript titled "BMI1 regulates PRC1 architecture and activity through homo- and hetero-oligomerization" by Gray et. al. investigates the structure of complex between BMI1-UBL domain and HD1 region of PHC2. The UBL domain adopts an expected ubiquitin-like structure, it binds a short 24 residue beta-hairpin of PHC2-HD1. The authors have found that the Bmi-UBL domain can also homo-oligomerize. The hetero- and homo-oligomerization are important for in H2A ubiquitination and clonogenic activity. This is a nice structure-function study. However, there are two major concerns regarding the paper. I have provided them below along with the minor issues.

Major Issues

1. The authors have claimed "point mutations in the BMI1 UBL domain which disrupt protein-protein interactions or homo-oligomerization of BMI1 impair the E3 ubiquitin ligase activity of the PRC1 complex". (Page 11, line 12)

Comments: It is clear that the hetero- and homo-oligomerization is important for PRC1's ability to ubiquitylate H2A. However, the paper does not provide a mechanistic understanding of the role of these interactions towards H2A ubiquitylation. Does it impact the catalytic activity, E2/E3 interaction, E3/substrate (H2A) interaction or something else? There are no experiments to implicate any of these.

Response

According to our model presented in Figure 4H, BMI1 homo-oligomerization and interaction with PHC proteins are needed for proper assembly of the PRC1 complex and efficient H2A ubiquitination in cells. To further support this model, we tested whether PHC proteins are needed for H2A ubiquitination in HeLa cells.

First, we found that PHC2 and PHC3 but not PHC1 are expressed in HeLa cells (based on www.proteinatlas.org). We therefore performed knockdown of PHC2, PHC3 and PHC2+PHC3 in HeLa and we found that simultaneous knockdown of both PHC2 and PHC3 resulted in very strong reduction of H2A ubiquitination (Figure S12). This experiment is consistent with effect of the mutation in BMI1 (R165E/H174E) that disrupts the interaction with PHC proteins resulting in decreased H2A ubiquitination. Together, these data demonstrate that disruption of protein-protein interactions within the PRC1 complex is sufficient to reduce H2A ubiquitination without directly affecting catalytic activity of RING domains of BMI1 and RING1A/1B.

We have added new data showing the effect of PHC2 + PHC3 knockdown on H2A ubiquitination in Figure S12 and revised text in the Discussion (page 15) to emphasize the need for an intact PRC1 complex for H2A ubiquitination activity.

2. The authors have claimed in discussion (Page 13 21) that "to our knowledge, our work represents a first example that requires a combination of X-ray crystallography and NMR data to determine the overall structure of a protein-protein complex". In their Materials and Methods the structure determination part states "NMR structures of PHC2 were combined with the X-ray crystal structure of BMI1 UBL domain and the entire complex was refined using Rosetta 45,46,67 constrained by NOE-derived distances restraints, dihedral angle restraints and BMI1 crystal structure coordinates"

Comments: These calculations by taking the NMR structure of one protein, x-ray structure of the other and forming the complex using intermolecular restraints is routinely done (Pruneda et. al., Mol cell, 2012). In fact, algorithms like HADDOCK are famous for that. The author's claim of novelty is not correct.

Response:

Our statement was indeed not precise and we are aware of many examples of using HADDOCK and Rosetta software for joint refinement using NMR and X-ray data. We have changed this text to emphasize that despite the collected X-ray crystallography data for BMI1-PHC2 complex, structure determination required additional restraints from NMR to determine complete structure of a complex.

The text on page 14 has been changed to:

“to our knowledge, our work represents a first example where X-ray crystallography data for a protein-protein complex were not sufficient for structure determination and NMR restraints were required to determine the overall structure of the complex.”

Minor Issues

1. Page 8, line 5. The authors have stated that *"likely these residues are involved in long range interactions stabilizing the PHC2-BMI1 complex."*

Comments: The chemical shift perturbations observed in 48-56 could be due to minor structural changes in that region caused by the impact of the binding of the 33-47 part. How do the authors speculate that this is due to long-range interactions?

Response:

Based on our structure, we expect that residues in this region, e.g. Glu48 in PHC2 are involved in the electrostatic interaction with Arg165 in BMI1. We have added following explanation to the text to clarify this (page 8):

“It is likely that these residues are involved in long range electrostatic interactions stabilizing the PHC2-BMI1 complex (e.g. interaction of PHC2 Glu48 with BMI1 Arg165).”

2. *In the designs of mutations to disrupt BMI1:PHC2 interface, only electrostatic interactions were targeted. Are the hydrophobic interactions dispensable?*

Response:

In the original submission of the manuscript we have presented the data showing mutations of R165 and H174 to glutamic acids in BMI1 to introduce electrostatic repulsion with PHC2. The side chains of R165 and H174 are solvent exposed and we expected that these mutations will minimally affect the structure of BMI1. However, the hydrophobic interactions are most likely very important for the binding. To probe this, we made an additional point mutation in BMI1, M170E, and found that this mutation significantly disrupts the interaction with PHC2 (~80-fold reduction in K_D relative to wild type BMI1). New binding data for BMI1 M170E- PHC2 interaction are presented in Figure S7C. We have updated the text on page 9 as follows to describe these data:

“To further probe whether hydrophobic contacts contribute to the BMI1-PHC2 interaction we introduced M170E point mutation in BMI1 and found that it significantly reduced the binding to PHC2 by ~80 fold ($K_D = 17 \pm 1.7$ uM) (Figure S7C).”

3. Page 10, line 1. *The K_d for I212E mutant does not match with the Figure 3C.*

Response:

We thank Reviewer for finding this typo. The correct value is 1.04 ± 0.25 uM and we correct that in the text.

4. Page 10 line 6. *The difference in line-broadening is not evident from the 2D plots. Please provide cross-sections.*

Response:

We have provided cross-sections in Supplementary Figure 8 to clearly show the broadening effect (Figure S8D).

5. *The authors have not presented statistics and validation of Rossetta generated structures.*

Response:

Statistics and validation of the structures generated by Rosetta have been provided in Table 2.

Reviewer #2 (Remarks to the Author):

The manuscript by Gray and colleagues provides a structural model for PCGF4/BMI1 interaction with HD1 domain in Polyhomeotic paralogs. It also describes, structurally the interaction between PCGF4/BMI1 surfaces responsible for a putative oligomerization.

The data are of interest to all researchers interested in Polycomb function and contributes to the disentanglement of biochemical complexity(es) of type I Polycomb Repressive Complexes.

I believe, however, that studies on the functional implications inferred from new structures should be reinforced/clarified.

For instance, binding of truncated BMI1 protein to PHC2 in Fig. 1B demonstrates that the deleted region (1-105 amino acids deleted) is dispensable for such an interaction. It would have been informative, however, to compare it with binding to full length BMI1. I'd guess a larger amount of PHC2 would be co-immunoprecipitated, considering the apparent low (in a transiently transfected 293 cell system) amount of PHC2 bound to truncated BMI1.

Response:

To address Reviewer's concern we have performed immunoprecipitation (IP) experiments in HEK293 cells. Results from Flag IP of full length BMI1 from HEK293 cells demonstrate interaction between PHC2 and full length wild type BMI1 but not with full length BMI1 R165E/H174E. Furthermore, deletion of residues 30-51 from PHC2 also abolished the interaction with full length BMI1. These new data are presented in Figure S1.

We have added text on pages 4-5 to reflect this new experiment:

"Further co-immunoprecipitation experiments confirmed the interaction of full length BMI1 with PHC2 (Figure S1)."

Q#2

Experiments in Fig 4: it can be understood that wild type and mutant BMI1 cDNAs are full length forms but it doesn't harm to clarify this. Also, it would have been convenient that cDNAs were tagged so that they could be distinguished from endogenous BMI1 protein. I would imagine that siRNA treatment (Fig. 4B) may have a transient effect on the endogenous protein and, if as I understand, plasmid transfection is in the absence of siRNA re-expression of endogenous BMI1 would be occurring.

Response

It is correct that BMI1 constructs used in these experiments were encoding full length protein with Flag-tag. We have clarified this in the legend of Figure 4B. To demonstrate overexpression of Flag-tagged BMI1 we have performed western blot with anti-Flag antibody and added these new data to Figure 4B.

This new western blot clearly demonstrates similar levels of expression of Flag-tagged wild type BMI1 and the three mutants.

Q#3

Although the differences in UbH2A are clear it would have been convenient that transfection efficiencies could be normalised. Also, importantly, western blot measuring global UbH2A levels corresponding to longer exposures will allow a better evaluation of the mutations investigated. A concern is that although decreased, UbH2A levels remain functionally significant.

Response

To address this concern we have provided longer exposure for UbH2A in the supplementary Figure S9. From these data, it is evident that both the double mutant R165E/H174E and point mutant I212E have significantly reduced UbH2A levels. To demonstrate that there is efficient transfection and similar levels of expression of Flag-BMI1 constructs we have performed additional western blot and probed for Flag. These new data are shown in Figure S9.

Q#4

That BMI1 variants have an impact on clonogenicity (no data for proliferation is given, despite the text says that proliferation is affected) may not necessarily be determined only by alteration, particularly if modest, of UbH2A levels. Unless powerful dominant effects are taking place, it has to be considered the presence of additional PCGF members acting as positive cofactors of RING1 E3 ligase.

Response

We appreciate the Reviewers' comment; to clarify that data are for clonogenicity we have modified the text throughout the manuscript and changed "BMI1 knockdown on proliferation" to "BMI1 knockdown on clonogenic potential."

We have also modified the text (page 12) to emphasize dominant negative effect possibly through the activity of PCGF members:

"... again supporting that R165E/H174E and I212E mutants function as dominant negatives (Figure 4F, G) possibly through activity of additional PCGF members"

Q#5

It would also have been desirable to discuss about the specificity of interactions of HD1 domains in PHC paralogs with PCGF2 (not studied here) and PCGF4. Surely, if PHC proteins bind BMI1 sequences present in all PCGF paralogs there is got to be other regions in these PCGF proteins that help discriminate interactions.

Response

The Reviewer asked a very important question and we decided to experimentally test whether PCGF2 (MEL18) interacts with PHC2. We expressed recombinant MEL18 UBL domain (construct equivalent to BMI1 UBL domain) and performed fluorescence polarization assay to test the binding to PHC2 fragment. We found that MEL18 also interacts with PHC2, although with a slightly reduced affinity when compared to BMI1. These new data are presented in Figure S11.

We have added Figure S11 and short paragraph in the Discussion (page 13) describing these results. We have also included short discussion on similarity of BMI1 UBL domain to other members of PCGF family, as follows:

"Sequence alignment of BMI1 homologs (PCGF family) shows that the UBL domain of BMI1 is most similar to MEL18 (PCGF2) with 60% sequence identity, while it differs significantly from the remaining homologs (Figure S11A,B). To assess whether the MEL18 UBL domain interacts with PHC2 we tested the binding of MEL18 residues 121-237 with PHC2₃₂₋₆₁ using the FP assay. We found that MEL18 UBL binds PHC2 with slightly reduced affinity ($K_D = 812 \pm 9$ nM) when compared to BMI1 (Figure S11C). Analysis of the BMI1-PHC2 structure revealed that all BMI1 residues involved in contacts with PHC2 are identical with MEL18 and strongly suggests very similar binding mode for MEL18-PHC2. Sequences of the remaining homologs of BMI1 differ significantly and most likely do not interact PHC proteins. "

Q#6

Interestingly, a mutation in the BMI1 oligomerization region has large impacts in UbH2A and clonogenicity assays: would it affect binding to PHC2?

Response:

We have already shown that point mutation I212E in BMI1 does not impair binding to PHC2 using the fluorescence polarization assay (Figure 3C). To further address Reviewer's question, we introduced I212E mutation to the full length BMI1 and performed immunoprecipitation with Myc-PHC2 in HEK293 cells. As expected, I212E mutation does not affect the BMI1-PHC2 interaction in cells. These new data are presented in Figure S1.

REVIEWERS' COMMENTS:

Reviewer #1 (Remarks to the Author):

The authors have responded to my queries and made appropriate changes. The manuscript will be interesting for the readers of Nature Communications. I recommend publication of the manuscript.

Reviewer #2 (Remarks to the Author):

As expected, the authors replied satisfactorily to points raised by the previous ms version.

I've only have a couple or comments on their reply:

-about Q#1, I realize the authors are rather more interested in making the point of how BMI1 R165E/H174E or PHC2 Δ 30-51 interfere with their association. However, as it was mentioned in my first comment, the interest of looking at full length BMI1-PHC2 CoIP was more to compare the extent of their association with that of truncated (Δ 1-105) BMI1. Neither of the legends to Fig. 1B or Suppl. 1 indicate amount of material (input or pulldown) loaded in the gels, making impossible to withdraw conclusions about these interactions.

-about Q#2, 3, I find confusing the lack of correlation between the intensities of FLAG and BMI1 antibodies in some lanes, because tagged BMI1 will also be detected always by anti-BMI1.

Nice piece work though, aimed to a better understanding of the intricacies of PRC1 complexes.

Response to the Reviewers

We thank both Reviewers for their time to review our manuscript and for providing valuable comments. The Reviewer #1 has recommended the manuscript for publication, while Reviewer #2 asked to clarify two points as specified below in details.

Reviewer 2 asked to clarify two more points:

-about Q#1, I realize the authors are rather more interested in making the point of how BMI1 R165E/H174E or PHC2 Δ 30-51 interfere with their association. However, as it was mentioned in my first comment, the interest of looking at full length BMI1-PHC2 CoIP was more to compare the extent of their association with that of truncated (Δ 1-105) BMI1. Neither of the legends to Fig. 1B or Suppl. 1 indicate amount of material (input or pulldown) loaded in the gels, making impossible to withdraw conclusions about these interactions.

Response:

In order to address the Reviewer's comment, we have performed dedicated experiment where we directly compared co-immunoprecipitation between Myc-tagged PHC2 and Flag-tagged full length BMI1 and truncated BMI1 (106-326 fragment of BMI1). Again, we confirmed that PHC2 interacts with both full length BMI1 and BMI1 106-326 fragment (see Supplementary Figure 1b). In both cases we used 500 μ g of whole cell lysates for co-IP and found that PHC2 interacts slightly weaker with full length BMI1 when compared to BMI1 106-326 fragment.

The N-terminal RING domain in full length BMI1 is most likely unfolded in the absence of Ring1B binding partner. It is likely that this unfolded RING domain partially masks the UBL domain leading to slightly weaker interaction with PHC2. The new data is included as Supplementary Figure 1b.

-about Q#2, 3, I find confusing the lack of correlation between the intensities of FLAG and BMI1 antibodies in some lanes, because tagged BMI1 will also be detected always by anti-BMI1.

Response:

We agree with the Reviewer that the intensities of bands for Flag and BMI1 antibodies should correlate. There is indeed slight difference in the intensities of bands between Flag and BMI1, particularly for the BMI1 R165E/H174E mutant. Most likely this results from technical imperfections of the western blot (such as antibody detection sensitivity). Importantly, we believe these small differences do not affect the conclusions we have drawn in the manuscript.

To further investigate this matter, we have performed new independent experiment (see Supplementary Figure below) and observed a good correlation between Flag and Bmi1 detection (see Supplementary Figure A). We have tested transfection efficiency using GFP and found very similar levels for all constructs (see Supplementary Figure B). Based on the western blot, we observed some variability in the expression levels between wild-type BMI1 and the mutants. Most importantly we have obtained a very

similar effect for BMI1 mutants on the ubiquitination levels of H2A to the results we report in the manuscript, fully supporting our initial findings.

Supplementary Figure. A. This is a new experiment to characterize Ub-H2A levels upon BMI1 knockdown followed by overexpression of BMI1 in HeLa cells. HeLa cells transfected with control or BMI1 3' UTR siRNA for 48 hours were transfected with plasmids encoding Flag-tagged full length wild-type BMI1 or mutants and analyzed by immunoblotting after 48 hours. **B.** Transfection control using GFP expression.

We hope that these new experiments address Reviewer's questions and our manuscript is suitable for publication.